# Strain-level detection of *Fusobacterium nucleatum* in colorectal cancer specimens by targeting the CRISPR–Cas region

Yumi Shimomura,[1] Yutaka Sugi,[1] Aiko Kume,[1] Wataru Tanaka,[1] Tsutomu Yoshihara,[2] Tetsuya Matsuura,[2] Yasuhiko Komiya,[2] Yusuke Ogata,[3] Wataru Suda,[3] Masahira Hattori,[3] Takuma Higurashi,[2] Atsushi Nakajima,[2] Mitsuharu Matsumoto[1]

**ABSTRACT**  *Fusobacterium nucleatum* is associated with colorectal cancer (CRC), and identical *F. nucleatum* strains seemed to be detected in 75% of patients who exhibited the presence of *F. nucleatum* in both CRC and saliva samples in our previous study; however, the validation of strain identity and the development of a method for strain-level discrimination are required for etiological studies. We confirmed that *F. nucleatum* isolates obtained from CRC and saliva samples derived from patients with CRC originated from their identical strains using whole-genome sequencing. We evaluated the hypervariable region of the clustered regularly interspaced short palindromic repeats (CRISPR) and CRISPR-associated (CRISPR–Cas) system in *F. nucleatum* strains isolated from the CRC and saliva specimens of CRC patients to develop a method for genotyping this bacterium at the strain level. The developed method consisted of two simple PCR steps to amplify the different lengths and sequences of the CRISPR–Cas regions from *F. nucleatum* strains using specific but common primers. This method could successfully detect identical strains present in both cryopreserved CRC samples and saliva obtained from the same CRC patient. Dynamic monitoring of *F. nucleatum* strains in saliva obtained from patients with colorectal adenoma before and after oral care showed interindividual variability in *F. nucleatum* strains during oral care. This study provided a simple and rapid method for comprehensively identifying *F. nucleatum* at the strain level in clinical samples, leading to a paradigm shift in CRC research, such as the investigation of pathogenic *F. nucleatum* strains and monitoring of pathogenic strains in clinical trials for preventing CRC recurrence. (This study was registered in the UMIN Clinical Trials Registry under ID UMIN000016229.)

**IMPORTANCE**  *Fusobacterium nucleatum* is one of the predominant oral bacteria in humans. However, this bacterium is enriched in colorectal cancer (CRC) tissues and may be involved in CRC development. Our previous research suggested that *F. nucleatum* is present in CRC tissues originating from the oral cavity using a traditional strain-typing method [arbitrarily primed polymerase chain reaction (AP-PCR)]. First, using whole-genome sequencing, this study confirmed an exemplary similarity between the oral and tumoral strains derived from each patient with CRC. Second, we successfully developed a method to genotype this bacterium at the strain level, targeting the clustered regularly interspaced short palindromic repeats (CRISPR) and CRISPR-associated system, which is hypervariable (defined as *F. nucleatum*-strain genotyping PCR). This method can identify *F. nucleatum* strains in cryopreserved samples and is significantly superior to traditional AP-PCR, which can only be performed on isolates. The new methods have great potential for application in etiological studies of *F. nucleatum* in CRC.

**KEYWORDS**  *Fusobacterium nucleatum*, colorectal cancer, CRISPR–Cas system, polymerase chain reaction, strain level, saliva

Address correspondence to Mitsuharu Matsumoto, m-matumoto@meito.co.jp.

Yumi Shimomura and Yutaka Sugi contributed equally to this article. Author order was determined in order of increasing seniority.

The authors declare no conflict of interest.

See the funding table on p. 16.

According to the Global Cancer Statistics 2018 using the GLOBOCAN 2018, over 1.8 million new colorectal cancer (CRC) cases and 881,000 deaths are estimated to have occurred in 2018, accounting for about 1 in 10 cancer cases and deaths, and CRC ranks third in terms of incidence but second in terms of mortality (1). *Fusobacterium nucleatum* predominately colonizes the oral environment of humans and is pathogenically responsible for periodontitis (2). *F. nucleatum* is abundant in human CRC tissue relative to adjacent normal colonic tissue (3–6), and this abundance has been positively correlated to disease stage and a poorer patient survival rate (7–9). Some tested strains of this bacterium induce the reduction of T cell-mediated anti-tumor immune response (5), mutation of the CpG island methylator phenotype, and microsatellite instability (10–12). Bacterial etiological approaches have shown that *F. nucleatum* exacerbates CRC by adhesion or invasion via the outer membrane proteins, *Fusobacterium* adhesion A (FadA) (13), and fibroblast activation protein 2 (Fap2) (14).

However, in the past decade, the pathological mechanism and preventive strategy of CRC have not been elucidated. Studies investigating the relationship between CRC and *F. nucleatum* have been performed using metagenomics or 16S rRNA gene amplicon sequencing based on next-generation sequencing (NGS) technologies, but these NGS-based studies have limitations in terms of the resolution at the strain level. In our previous study, identical strains were detected in 75% of patients who showed the presence of *F. nucleatum* in both CRC and saliva specimens, suggesting that there is a high possibility that *F. nucleatum* in CRC originates in the oral cavity. In addition, only one to two *F. nucleatum* strains out of three to eight strains in saliva were present in CRC, suggesting the involvement of the strain-level pathogenicity of this bacterium in the development of CRC (15). In fact, differences in different strains of the same species are crucial for several tasks in the field of pathogenic bacteria because the unit of microbial action is a strain, not a species (16). The difference in Caco-2 cell invasion (17), CRC cell line binding (14), natural killer cell cytotoxicity of tumor via Fap2 (18), and adhesion to and activation of lymphocytes (19) at the strain level supports this.

Bacterial strain typing methods can be based on genomic data (20). These analyses commonly require live bacteria isolated by culturing. However, culturing has several disadvantages, including the need to cultivate from specimen within the sampling day before bacterial death, the long processing time required for incubation and purification of isolates obtained from agar, culture selection bias depending on experimental skill, and the difficulty of selecting minor strains among predominant strains. Therefore, for strain-level studies, frozen specimens are not acceptable and high-throughput analyses are impractical.

Clustered regularly interspaced short palindromic repeats (CRISPR) and CRISPR-associated (Cas) (CRISPR–Cas) system comprises a novel mechanism of bacterial resistance to phage infection (21), which is widespread in the genomes of approximately 50% of bacteria and 90% of archaea (22). CRISPR loci contain short, partially palindromic DNA repeats that occur at regular intervals and form loci with alternate repeated elements (CRISPR repeats) and variable sequences (CRISPR spacers) that are homologous to exogenous genetic elements, including viruses and plasmids (23). This preserves the history of infection, and differences in invasive exogenous genetic sequence are reflected in the different lengths and sequences of CRISPR, resulting in different hypervariable regions for each strain. CRISPR-based genotyping has been studied in several bacterial species including *Mycobacterium tuberculosis* (24, 25), *Campylobacter jejuni* (26), and *Streptococcus pyogenes* (27).

Here, we investigated the use of CRISPR–Cas system-based PCR for bacterial strain typing of *F. nucleatum* and demonstrated the effectiveness of this method in detecting strains of *F. nucleatum* in the CRC specimen and saliva obtained from CRC patients.

## RESULTS

### Validation of identical *F. nucleatum* strain obtained from both CRC and saliva using whole-genome sequencing

Randomly selected five pairs of isolates obtained from CRC and saliva of each CRC patient, which were suggested as identical strains by the arbitrarily primed polymerase chain reaction (AP-PCR) in our previous study (15), were analyzed using whole-genome sequencing [Isolate C-A3 in CRC and Isolate S-A3 in saliva are Strain A3 derived from Patient D; Isolate C-P4 in CRC and Isolate S-P4 in saliva are Strain P4 derived from Patient E; Isolate C-V3 in CRC and Isolate S-V3 in saliva are Strain V3 derived from Patient F; Isolate C-P10 in CRC and Isolate S-P10 in saliva are Strain P10 derived from Patient G; Isolate C-P11 in CRC and Isolate S-P11 in saliva are Strain P11 derived from Patient H in our previous paper (15)]. Using pairwise whole-genome comparison for each pair, the percentage of single nucleotide variants (SNVs) (the number of SNVs/genome size) between paired strains was 0.00036%–0.0038%, which was significantly lower than those in between non-paired strains (1.228%–3.159%; mean, 2.374%) (Table S1), demonstrating that these pairs of isolates are originated from identical strains.

### CRISPR–Cas system of *Fusobacterium nucleatum*

Based on the Cas protein sequences and gene structures, the CRISPR–Cas system can be classified into several types (28–30). By *in silico* analyses, the CRISPR–Cas system possessed by 26 strains of *F. nucleatum* in the NCBI database was classified into three subtypes, Types I-B, II-A, and III-A. We further categorized the Type I-B into two subtypes, Type I-B1 and Type I-B2, based on the similarity of *Cas* genes as well as the position of this system conserved on the genome (Fig. 1). Types I-B1 and I-B2 are similar in the genomic positions of their *Cas* genes. In addition, their *Cas1* genes and Cas1 proteins are 66% and 44% similar in nucleotide sequences and amino acid sequences, respectively. Meanwhile, the aligned sequence of the *Cas* genes was highly conserved (Fig. 1). The presence or absence of these CRISPR–Cas systems for each strain in the database is illustrated in the Table S2.

### PCR for detection of *F. nucleatum* CRISPR-associated regions (*F. nucleatum*-strain genotyping PCR)

Primer sets for CRISPR-associated regions are given in Table S3; these primer sets were designed based on the DNA sequences of 26 *F. nucleatum* strains obtained from the NCBI database (Table S2) and *Cas2* sequences of *F. nucleatum* clinical isolates (accession numbers LC583793–LC583797, LC585884–LC585886, and LC592376–LC592400) obtained from our previous study (15). In the preliminary experiment using saliva, the PCR amplicon generated by CRISPR-associated region targeting PCR was undetectable in several cases; alternatively, in some cases, non-CRISPR-associated regions were amplified. Hence, we designed primer sets for nested PCR (Table S3) and successfully increased the CRISPR detectability using the two-step PCR comprising CRISPR-associated regions targeting PCR (first PCR) and nested PCR (second PCR) (defined as *F. nucleatum*-strain typing PCR).

*F. nucleatum*-strain genotyping PCR was performed to confirm the amplification of the CRISPR-associated regions of *F. nucleatum*-type strains. The result showed that these primer sets detected distinct bands for Type I-B1 and Type III-A, Type I-B2, Type I-B2, and Type II-A from *F. nucleatum* subsp. *animalis* JCM11025[T], *F. nucleatum* subsp. *nucleatum* JCM8532[T], *F. nucleatum* subsp. *polymorphum* JCM12990[T], and *F. nucleatum* subsp. *vincentii* JCM11023[T], respectively (Fig. 2a). Furthermore, the amplicon size of each band was as expected (Table S4). All sequences of these bands were 100% matched (E-value was 0.0, the highest alignment score) to the original genomic sequences (Table S5), and palindromic repeat sequences were found in PCR products obtained from three strains, demonstrating that the targeted CRISPR-associated regions were accurately amplified by this PCR procedure (Fig. 2b; Fig. S1).

## Type I-B1

Representative: *Fusobacterium nucleatum* subsp. *animalis* strain 7_1
(CP007062, FSDG_02374 - FSDG_04051)

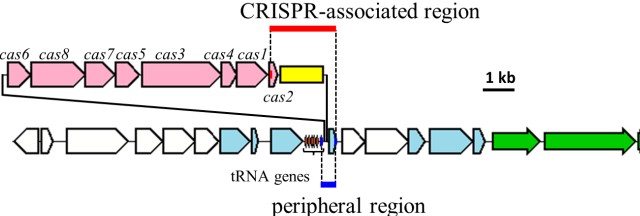

## Type I-B2

Representative: *Fusobacterium nucleatum* subsp. *nucleatum* strain ATCC 23726
(CP028109, C4N14_06130 - C4N14_06285)

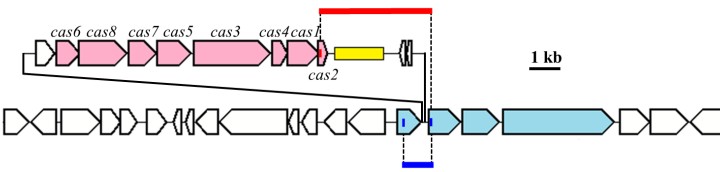

## Type II-A

Representative: *Fusobacterium nucleatum* subsp. *vincentii* strain 3_1_27
(CP007064, HMPREF0405_01059 – HMPREF0405_01036 (complementally))

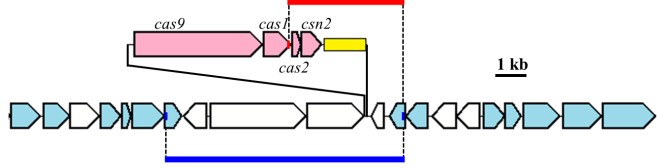

## Type III-A

Representative: *Fusobacterium nucleatum* subsp. *vincentii* strain 4_1_13
(NZ_KQ235735, FSCG_RS03545 – FSCG_RS03690 (complementally))

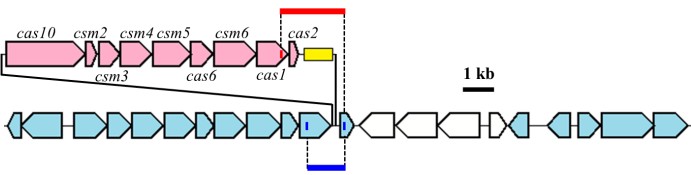

**FIG 1**  Schematic gene arrangement and positions of primers designed for the amplification of *F. nucleatum* CRISPR–Cas system. *Cas* gene, CRISPR region, conserved genes, and non-conserved genes are in pink, yellow, sky blue, and white, respectively. Small orange arrows indicate tRNA genes, and green arrows indicate rRNA genes. The blue vertical line in conserved genes indicates the primer positions that amplify the peripheral region of the CRISPR–Cas system (blue horizontal line). The red vertical line in *Cas* gene indicates the forward primer position that amplifies the CRISPR region; the reverse primer for this region is common to the reverse primer for peripheral region of the CRISPR–Cas system. If there is a CRISPR region, the red horizontal line region is amplified. Representative strains are not type strains.

## Strain-level discrimination of clinical isolates obtained from CRC patients

To investigate whether *F. nucleatum*-strain genotyping PCR is effective for strain-level discrimination of clinical strains, we performed PCR for genomic DNAs of the representative isolates of 23 strains (Fig. S2) obtained from five saliva specimens randomly selected

## (a)

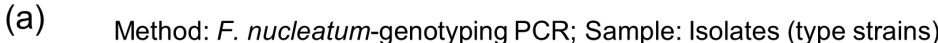

Method: *F. nucleatum*-genotyping PCR; Sample: Isolates (type strains)

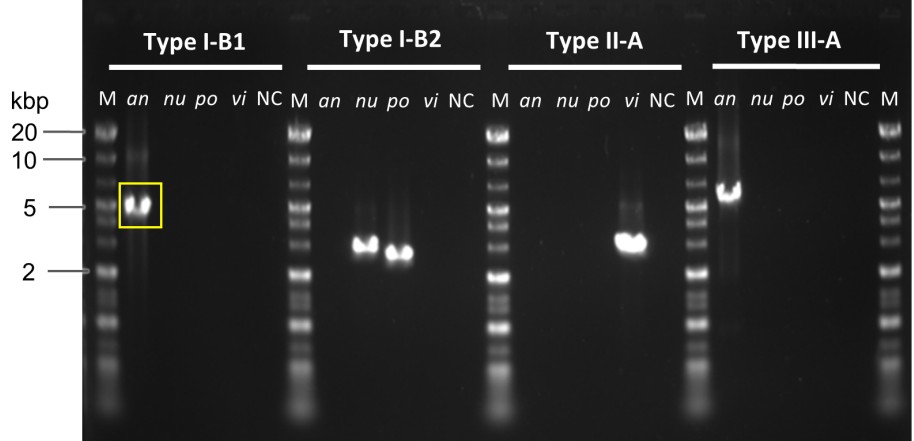

## (b)

### *F. nucleatum* subsp. *animalis* JCM11025<sup>T</sup>, Type I-B1 5′-end

```
AATTAAAAAG CATAATAGTT GAAAATGAAG ATTTCGTTTG TATTTTAAAA TCAATAAACC CTGATGTTTT
TGGAGAAGAA ACTTTGGGAA ACCCAACTCC AAATGGAGAA AATATATTTC TATAATTTAA AAATTTTCCC
AAGCAATATT CTAAAAAACT ATTTAAACTT TCCAAAATAT GATTATTTTA GAAAAATTAA GTAAAAAATA
GAATTGCTTG GGAAAAACGT ATAAAAAATA CTTGTAATAT TGAAAAAAAT AAGGTATTAT AAATTTATAA
AGCTTGAAAA AATAGCTATG AACTATAAAC TTGAAAAGTT TTGAAATATA TCATAATCTT GCTTAATTAC
GTTATATTTA ATGAACTATA AACTTGAAAA GTTTTGAAAT CTAGCTTTAC AGGTGAGATC TGTTAGGTTA
TCTAATATGA ACTATAAACT TGAAAAGTTT TGAAATAATA GATCCTGTAA AAGAAGTTAA TGCTTCTATT
TTATGAACTA TAAACTTGAA AAGTTTTGAA ATTGTATTGT AGCTCCTGTT AAAGCCTTAA CTTTAAATGA
ACTATAAACT TGAAAAGTTT TGAAATTATT TTTTCTTTA
```

### *F. nucleatum* subsp. *animalis* JCM11025<sup>T</sup>, Type I-B1 3′-end (Reverse complement)

```
TTTTGAAATA AAGGAGAAAA CACACTAGGT ATTAATCAAA GCTATATGAA CTGTAAACTT GAAAAGTTTT
GAAATAACAA ATATAGTTTC TCTTTATTTG CACAGTTTAT ATGAACTGTA AACTTGAAAA GTTTTGAAAT
TACCTTAGCA GATAAAATTT TTATGCATAT CATGAAATGA ACTATAAACT TGAAAAGTTT TGAAATCAAG
CTACGGCATA CAGGACTATA AATGAAATTA AATGAACTGT AAACTTAAAA AATTTTGTTG TTAAAAAGTA
TTTAGAAAAA TATTAGGTAG ATACAAAACA AAATTAAGGT AAATAATTTA TGAATTGTAA AAAAATAAAG
AAAAACTAAT AAGTTGAATA TATTAATTTT TCCTATGAAG AAGTAGTAAA AATGGAAAAA TAACCAAACT
CATTTAATTG GAGTCAAATA TTATATAACA TACTAGTATA GAAATGAAAT ATGTTAGAAA GAAAAGAATT
ATAGATTTGA TAAAACAATT AAAAATTGTT GAAACAAGGA GGAAATTATG AAAAAATTTT CTATTCATGG
AACAGAAGAG
```

**FIG 2** Amplification of the targeted CRISPR region of *F. nucleatum*-type strains by *F. nucleatum*-strain genotyping PCR. (a) Electrophoresis pattern of PCR products of *F. nucleatum*-type strains. M, DNA ladder marker; an, *F. nucleatum* subsp. *animalis*; nu, *F. nucleatum* subsp. *nucleatum*; po, *F. nucleatum* subsp. *polymorphum*; vi, *F. nucleatum* subsp. *vincentii*; NC, negative control. The size of the marker fragments, from top to bottom, was 20, 10, 7.0, 5.0, 4.0, 3.0, 2.0, 1.5, 1.3, 1.0, 0.7, 0.5, 0.4, 0.3, 0.2, and 0.1 kbp; the intensified fragments are underlined. (b) DNA sequences of the 5′- and 3′-ends of PCR amplicon of *F. nucleatum* subsp. *animalis* Type I-B1 [yellow square in (a)] are shown and were used for the analysis. Repeat sequences are detected by CRISPRCasFinder and written in red. DNA sequences of other PCR amplicons are in Fig. S1.

from CRC patients (Patients D, E, F, G, and M) in our previous study (15), wherein their genomic DNAs were used as the PCR templates. All strains, excluding the strains lacking CRISPR-associated regions, generated distinguishable and unique bands (Fig. 3). The top panel (Fig. 3) displays the pattern of the bands of different strains generated by arbitrarily primed PCR (AP-PCR), which is a simple electrophoresis-based strain typing method applicable to various bacteria, including *F. nucleatum* (20, 31). For example, in Patient D, strain-specific amplicons and AP-PCR patterns were observed in each of the five strains (A2, A3, A4, A5, and P3). One strain (V1) lacked amplification in any subtypes of the CRISPR-associated regions. Primer sets for the peripheral region of the CRISPR–Cas system were also designed (Table S3; Fig. 1, blue horizontal line) to confirm the absence of all CRISPR–Cas system of the seven strains (V1, P5, V2, V3, V4, V5, and A12) that lacked amplification by the *F. nucleatum*-strain genotyping PCR (Fig. 3). The peripheral regions

## Method: AP-PCR or *F. nucleatum*-genotyping PCR; Sample: Isolates
### The pattern of CRISPR-associated regions possession of each strain

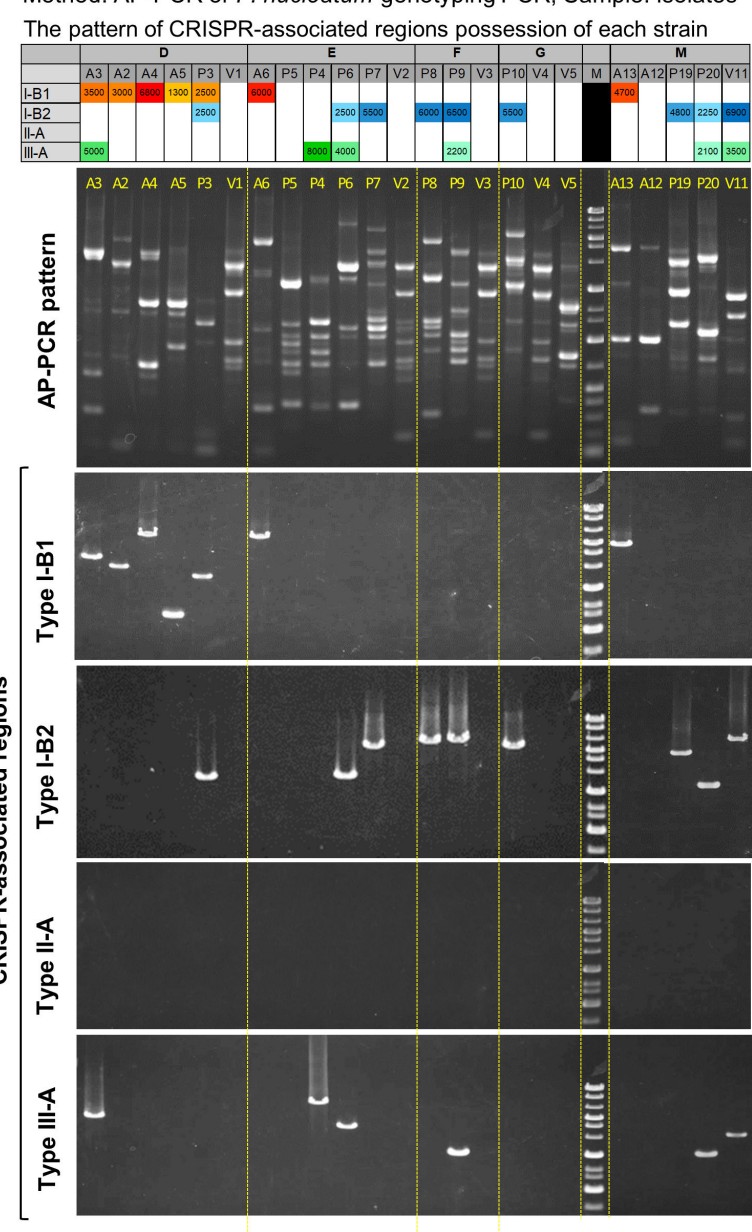

**FIG 3** Strain-level discrimination of *F. nucleatum* isolated from saliva of CRC patients by CRISPR-associated region-targeted PCR. Representative isolates of *F. nucleatum* strains derived from saliva of CRC patients (Fig. S2) were analyzed. The top table shows the pattern of the CRISPR-associated regions of each strain. The numbers in the table are PCR product sizes shown in light (small sized) to darker (large sized) colors. In Patient D, five strains (A2, A3, A4, A5, and P3) were distinguished, although one strain (V1) lacked the CRISPR-associated regions. In Patient E, four strains (A6, P4, P6, and P7) were distinguished, although two strains (P5 and V2) lacked the CRISPR-associated regions. In Patient F, two strains (P8 and P9) were distinguished, although one strain (V3) lacked the CRISPR regions. In Patient G, one strain (P10) was distinguished from each strain, although two strains (V4 and V5) lacked the CRISPR regions. In Patient M, four strains (A12, A13, P19, and P20) were distinguished, although one strain (V11) lacked the CRISPR regions.

of any type of CRISPR–Cas system were confirmed by this PCR; however, the amplified size was smaller than the predicted size based on genome information, indicating that these strains have peripheral regions but do not possess the CRISPR–Cas system.

Additionally, all isolates obtained from saliva of one CRC patient (Patient F) (19 isolates classified to four strains based on the AP-PCR patterns) were analyzed by *F. nucleatum*-strain genotyping PCR (Fig. S3). As a result, the same-size amplicon was detected from all isolates identified as identical strains by AP-PCR, excluding strain V3 that did not have all the CRISPR regions. These were confirmed as identical strains by this PCR analysis.

Furthermore, we analyzed paired isolates, which were confirmed to be originated from the identical strain using whole-genome sequencing, by *F. nucleatum*-strain genotyping PCR (Fig. S4) and compared with PCR amplicon size estimated by genome sequences (Table S6). Subsequently, strain-specific amplicons with the same size as calculated from genome sequences were observed in each of the pairs, excluding pair of Isolates C-V3 and S-V3 that did not have all the CRISPR regions in genome sequence (Fig. S4). Interestingly, we observed a difference of approximately 400 bp in PCR amplicons between Isolates C-P10 and S-P10, which originated from the identical strain using whole-genome sequencing, by both the *F. nucleatum*-strain genotyping PCR and the calculation using genome sequences. This suggests that this strain has relocated its habitat to oral or CRC, resulting in a different phage infection history.

## Detection of *F. nucleatum* at the strain level by *F. nucleatum*-strain genotyping PCR from the specimen

We attempted to detect PCR products for each strain in the mixed genomic DNA of multiple strains because each CRC specimen or saliva sample contains multiple *F. nucleatum* strains, according to our previous study (15). First, *F. nucleatum*-strain genotyping PCR was performed using both single genomic DNA and mixtures of the same amount of each single genomic DNA as templates (Fig. 4a). PCR products corresponding to the CRISPR region from each strain were amplified separately and were approximately equal in density. Next, we detected *F. nucleatum* strains from saliva of CRC patients. Three saliva-derived DNA samples (obtained from Patients D, E, and M), from which we successfully extracted genomic DNA in our previous study (15), were analyzed through *F. nucleatum*-strain genotyping PCR. PCR amplicons from isolates were detected in all saliva-derived genomic DNA (Patient M in Fig. 4b; Patients D and E in Fig. S5). Interestingly, among the saliva-derived amplicons, there were several PCR products that were not found in isolated strains, indicating that this PCR can detect strains that could not be isolated by the culture method.

## Application for the survey of the identical strain presenting in paired CRC tissues and saliva

For clinical application, to confirm whether this method can detect identical strains of *F. nucleatum* present in both the CRC tissues and saliva, *F. nucleatum*-strain genotyping PCR was performed using total DNA extracted from both the CRC specimen sample and saliva obtained from three patients with CRC. We analyzed three additional patients (patient IDs O, P, and Q). From two patients (IDs O and Q), identical strains of *F. nucleatum* were detected in both CRC specimen and saliva by culture- and AP-PCR-dependent methods (Fig. S6 and S7). The *F. nucleatum*-strain genotyping PCR analysis detected identical strains from both specimens obtained from these patients (Fig. 5a). In addition, the size of each PCR amplicon was consistent with that of the PCR amplicon from each of the single genomic DNAs of the isolates, which were obtained from the same specimens by the culture method and identified as identical strains by AP-PCR. In other words, we succeeded in detecting the presence of identical strains present in both CRC specimens and saliva obtained from patients with CRC by *F. nucleatum*-strain genotyping PCR. Furthermore, to confirm whether these same-size amplicons are derived from the CRISPR-associated regions of identical *F. nucleatum* strains, the base sequences of these bands were analyzed. All the sequences of these bands were confirmed as CRISPR-associated regions of *F. nucleatum* by BLAST analysis and were almost 100% matched to each other (Fig. S8 and S9), indicating that *F. nucleatum*-strain genotyping PCR can detect

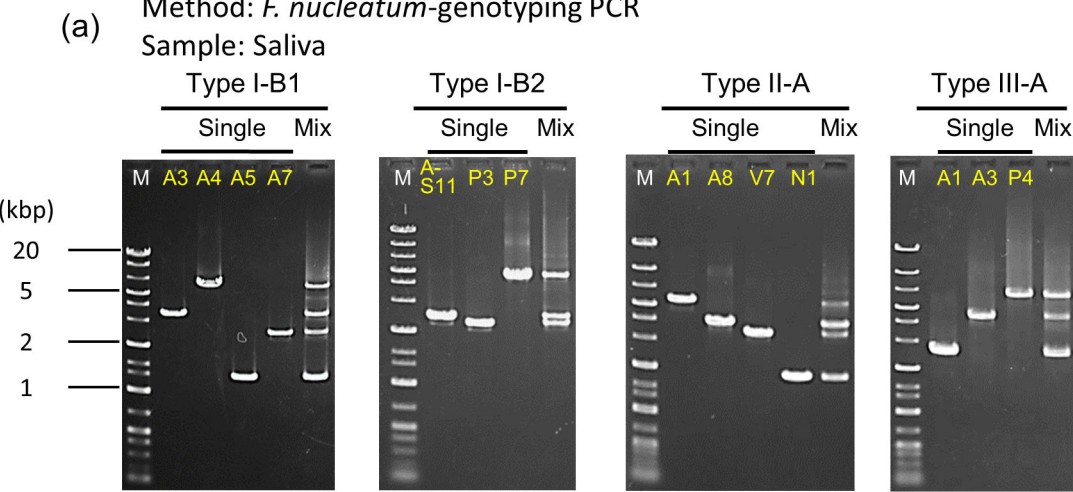

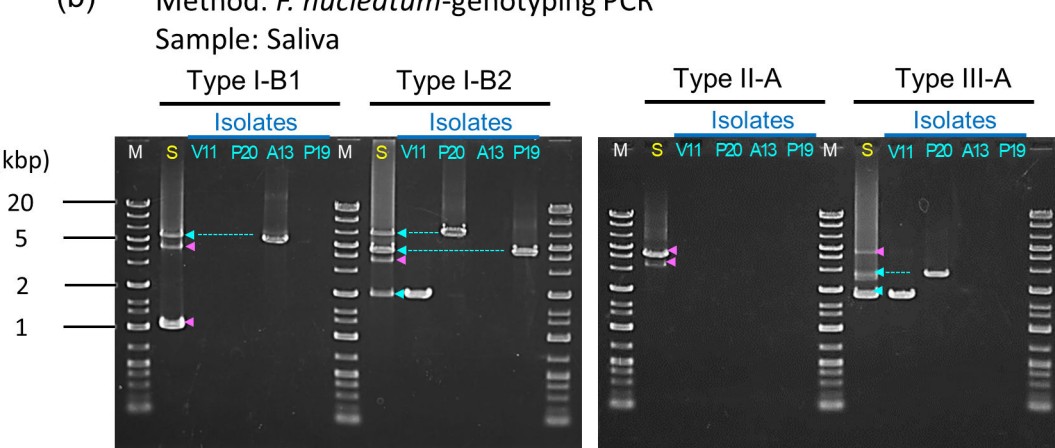

**FIG 4** Detection of *F. nucleatum* strains from saliva of CRC patients by the *F. nucleatum*-strain genotyping PCR. (a) Detection of PCR amplicons from single and mixed multiple genomic DNA of *F. nucleatum* strains. From left to right, *F. nucleatum* subsp. *animalis* A3, A4, A5, A7 (Type I-B1); *F. nucleatum* subsp. *polymorphum* A-S11, P3, P7 (Type I-B2); *F. nucleatum* subsp. *animalis* A1, A8, *F. nucleatum* subsp. *vincentii* V7, *F. nucleatum* subsp. *nucleatum* N1 (Type II-A); *F. nucleatum* subsp. *animalis* A1, A3, *F. nucleatum* subsp. *polymorphum* P4 (Type III-A). Origins of many isolates are shown in Fig. S2. Strains A7, A8, and V7, A-S11, and A1, which are not in Fig. S2, were isolated from saliva of Patient H, saliva of Patient A, and CRC of Patient C, respectively, in our previous study (15). (b) Single genomic DNA of isolate or total genomic DNA extracted from saliva (the origin of the isolates) was used as a template (Patient M). The lane of total saliva (left, "S" in yellow) contains the bands identified in the isolates (blue arrowhead) and the amplicons derived from other non-isolates (pink arrowhead). Data are not available for strain V2 of Patient E and A12 of Patient M because these do not have all CRISPR types (see Fig. 3).

identical strains present in both cryopreserved CRC specimens and saliva obtained from the same CRC patient. Analyzing the DNA sequences of the other two amplicons detected in saliva revealed that they were also CRISPR-associated regions derived from other *Fusobacterium* strains (Fig. S10). In contrast, for Patient Q, identical strains were not detected in both CRC specimen and saliva by culture- and AP-PCR-dependent methods (Fig. S11). The *F. nucleatum*-strain genotyping PCR analysis also did not detect any identical strains from both specimens obtained from this patient (Fig. S12).

## Application for the dynamic monitoring of *F. nucleatum* strains during oral care in patients with colorectal adenoma and periodontitis

This technique was applied to monitor specific pathogenic strains of *F. nucleatum* during clinical treatment, such as during the preliminary observations of the patients with

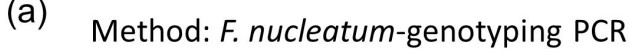

(a)

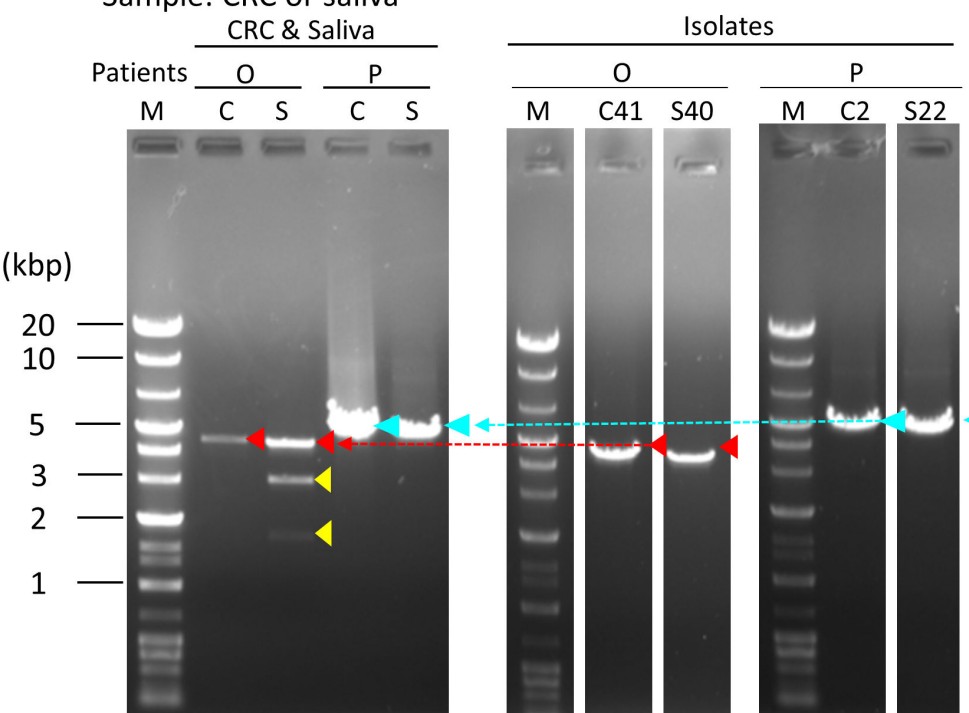

(b)

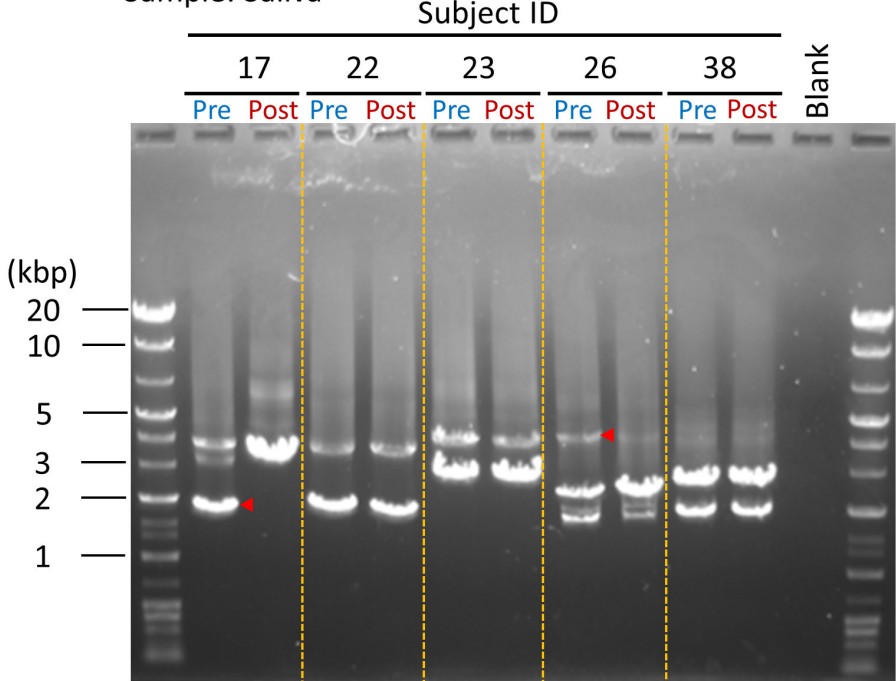

**FIG 5** Application of the *F. nucleatum*-strain genotyping PCR to clinical trials. (a) Detection of identical strains of *F. nucleatum* from both CRC specimen and saliva obtained from patients with CRC by *F. nucleatum*-strain genotyping PCR. In two of the three patients (O, P, and Q), the presence of identical strain was confirmed by both the culture method and *F. nucleatum*-strain genotyping PCR. Left panel, detection of strain-specific DNA amplicons from genomic DNA of the CRC tissue and saliva

**FIG 5** (Continued)

obtained from a patient with CRC (Type I-B2). A same-size amplicon (red or blue arrowhead) was detected from both the CRC tissue and saliva. Two amplicons (yellow arrowhead), which were detected only in the saliva, were also CRISPR-associated regions derived from other *Fusobacterium* strains. Right panel, PCR amplicons from a single genomic DNA of isolates, which were identified as identical *F. nucleatum* strains, obtained from the same specimens. In Patient Q, no identical strains were detected from both CRC specimen and saliva samples. Lanes of isolates were cut and pasted from the original gel pictures (Fig. S13) for clarity. M, marker (DNA ladder: 0.1 kbp to 20 kbp). (b) Monitoring of the dynamics of *F. nucleatum* strains in saliva specimen from patients with colorectal adenoma and periodontitis, who were given oral care. Genomic DNAs derived from saliva at pre- and post-oral care were analyzed by the *F. nucleatum*-strain genotyping PCR (Type II-A). PCR amplicons derived from strain removed or decreased substantially by the oral care are shown as red arrowheads.

colorectal adenoma. The dynamics of *F. nucleatum* strains in the saliva of patients with colorectal adenoma and periodontal disease, who were given oral care for approximately 3 months, were monitored via *F. nucleatum*-strain genotyping PCR. The results showed that one out of three strains detected in Subject 17 disappeared in the post sample, implying that this strain was markedly depleted in the abundance during oral care (Fig. 5b). In addition, a strain in Subject 26 decreased in the abundance substantially. These findings demonstrate that the present method can monitor alterations in oral *F. nucleatum* at the strain level during a specific period.

## DISCUSSION

Recent reports have shown that *F. nucleatum* is implicated in the development and progression of CRC (32, 33). A higher amount of tissue *F. nucleatum* was associated with a shorter CRC-specific survival in several independent studies (7, 9, 34). The relative abundance of *F. nucleatum* spp. was continuously elevated from intramucosal carcinoma to more advanced stages in a metagenomic analysis (35). *F. nucleatum* promotes carcinogenesis through surface adhesions such as FadA and Fap2. Specifically, Fap2 binds to a T-cell immunoreceptor with Ig and immunoreceptor tyrosine-based inhibitory motif domains (TIGIT) and impairs the function of CD4[+] T, CD8[+] T, and NK cells; moreover, it reduces cytotoxicity, resulting in tumor escape from immunosurveillance (18). This bacterium also modulates the resistance to chemotherapy by activating autophagy (36). Although *F. nucleatum* is fairly ubiquitous in the oral cavity, it is usually detected at low levels in the intestinal tract. Understanding how *F. nucleatum* strains and levels in the oral and intestinal tract affect CRC risk may provide suitable candidates for interventions focused on *F. nucleatum* modulation (32).

Metagenomics has facilitated understanding of how the gut microbiome at the bacterial phylum, class, order, family, genus, and species levels differs between individuals and between illnesses both with and without symptoms. However, there are many contradictory results between laboratories, possibly because of microbial genetic variations that perform different functions within the same species. Therefore, strain-level analysis may help resolve these contradictions and contribute to research about pathogen–host interaction (16). It is also conceivable that strain-level analysis uncovers novel microbiome-associated interactions, including the relationship between *F. nucleatum* and CRC development. Previously, we found that a few strains of *F. nucleatum* detected in the oral cavity were also found in CRC tissues (15). Furthermore, in this study, pairwise whole-genome comparison confirmed that specific isolates in CRC tissue were identical to specific isolates in saliva samples, demonstrating that at least part of patients with CRC have strains of *F. nucleatum* originated from their identical strains in their CRC and oral cavity. Therefore, we suggest that specific oral *F. nucleatum* strains, which may differ in key characteristics (such as enzyme or toxin expression), are involved in the development of CRC. In fact, there are differences in the presence of Fap2 and the adhesion rate to CRC cells at the strain level of *F. nucleatum* (14), further suggesting strain-specific pathogenicity in *F. nucleatum*. Accordingly, a strain-level approach could clarify the relationship between CRC and *F. nucleatum* and help in developing a

novel strategy for the prevention of recurrence after removal of CRC or suppression of carcinogenesis from a polyp.

Isolating live bacteria from a biological sample using the culture method is essential for the study of bacteria at the strain level. However, culturing has a number of disadvantages: (i) time constraints due to the need to culture immediately after sampling and before death of bacteria and the time spent in cultivation and purification and (ii) bias because of skill proficiency, colony selection, and antibiotic sensitivity with selective agar and the presence of undetectable bacteria buried in colonies of dominant bacteria. These constraints and biases result in low reproducibility and may be difficult to apply to the medical setting. In contrast, a culture-independent metagenomics approach has promisingly improved enumeration of microbial species in clinical samples at the strain level (16). However, this approach is laborious and costly and does not always guarantee the targeted analysis of certain species, particularly low-abundant species and strains in low-biomass samples. Hence, we developed a rapid PCR-based method targeting the CRISPR–Cas system in *F. nucleatum* strains, particularly those in cryopreserved specimens widely used in clinical settings. Our method detected almost all PCR amplicons from the strains isolated from saliva, which were also present in the PCR products from the same cryopreserved saliva, demonstrating that this method can be used for distinction of *F. nucleatum* at the strain level in a specimen. In addition, among the PCR products obtained from saliva, PCR amplicons that were not derived from isolated strains were detected, indicating that this method more sensitively and extensively identifies the strains than the culture method. In this study, we analyzed genomic DNA extracted from both CRC and saliva samples from three patients with CRC (patient IDs: O, P, and Q) using *F. nucleatum*-strain genotyping PCR. Subsequently, we detected identical *F. nucleatum* strains in both CRC and saliva samples in 66.7% (2/3) of the patients. However, this number of patients is too small, and further studies are needed.

These results indicate that the developed method is a powerful means of elucidating the relationship between *F. nucleatum* and CRC, facilitating prevention of the recurrence of CRC and the suppression of deterioration of adenoma (Fig. 6). For diagnosing CRC, CRC tissues from suspected patients are collected via endoscopy, and saliva and plaque specimens are collected from the oral cavity of patients and immediately frozen at −80°C. The genomic DNA is extracted from each specimen and subjected to PCR typing with our method. Detection of an identical strain in both specimens (CRC tissue and saliva) implies that this strain may be an oral pathobiont promoting CRC development. Application of this method to an increased number of cases will allow us to precisely estimate the prevalence of patients with CRC carrying the identical *F. nucleatum* strain in both oral cavity and CRC tissue, leading to a paradigm shift in CRC research. As the abundance of *F. nucleatum* is negatively correlated with life expectancy and CRC prognosis (9, 37), depletion of pathogenic *F. nucleatum* strains via appropriate oral care may prevent CRC recurrence. This method will be of great use to evaluate the abundance of pathogenic strains in clinical trials in a short time. Furthermore, this method can be applied to assess the effectiveness of oral care and risk management of CRC recurrence in regular clinical follow-up using saliva.

A limitation of this method is that *F. nucleatum* strains that do not possess the CRISPR-associated region are not detected. Based on our data and the sequences deposited in the NCBI database, the prevalence rate of the CRISPR-associated region in *F. nucleatum* was estimated to be 79.6% (39 strains out of 49 strains); therefore, approximately 20% of strains would be undetected by this method. In addition, it is difficult to determine whether two strains are identical based on PCR amplicons of similar length in a sample. To confirm whether the strains are identical, DNA sequencing of the PCR products is required. In the samples analyzed in this study, we observed amplicons with undistinguishable size in Patient P. These amplicons corresponded to three isolates, i.e., C10b, C2, and S22. The AP-PCR pattern of C10b differed from that of C2 and S22. DNA sequencing of the amplicons showed that C10b amplicon had a sequence identical to that of C2 and S22 amplicons (Fig. S14). This result suggests that *F. nucleatum*-strain genotyping PCR is a

## (a) Sampling and preservation

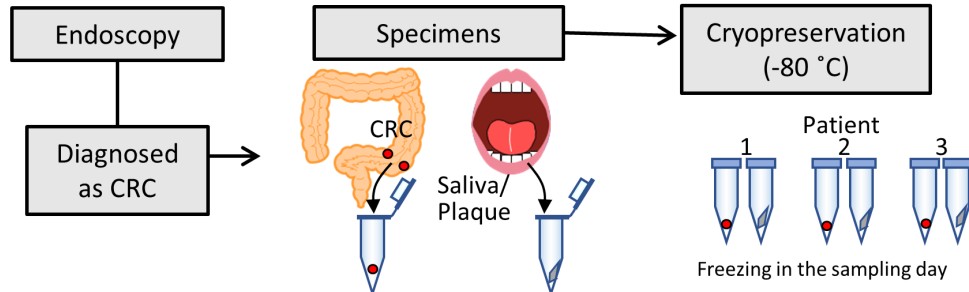

## (b) Procedure, interpretation, and application

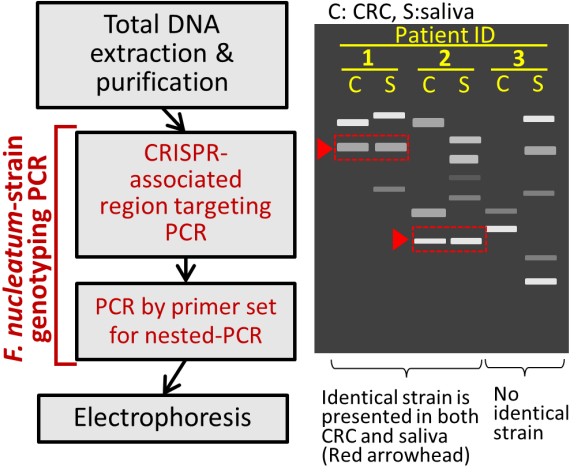

**Interpretation**
- Each amplicon (band) indicates the presence of one strain
- Same size amplicon (band) means identical strain
- Therefore, detection of same size amplicon from both CRC and saliva, indicating that the identical strain is present in both CRC and saliva (red arrowhead).

**Application**
**Distribution of identical *F. nucleatum* strains in paired CRC and saliva**
- Survey of population of patients with the identical strain is present in paired CRC and saliva
- Percentage of CRC caused by oral *F. nucleatum*

**Prevention of recurrence**
- Survey of relationship between recurrence and presence of identical *F. nucleatum*-strain
- Evaluation of oral care on recurrence of CRC
- Monitoring of oral *F. nucleatum* strains for risk management after CRC resection

## (C) Example: Dynamic monitoring of the identical strain in both CRC and saliva during clinical follow-up

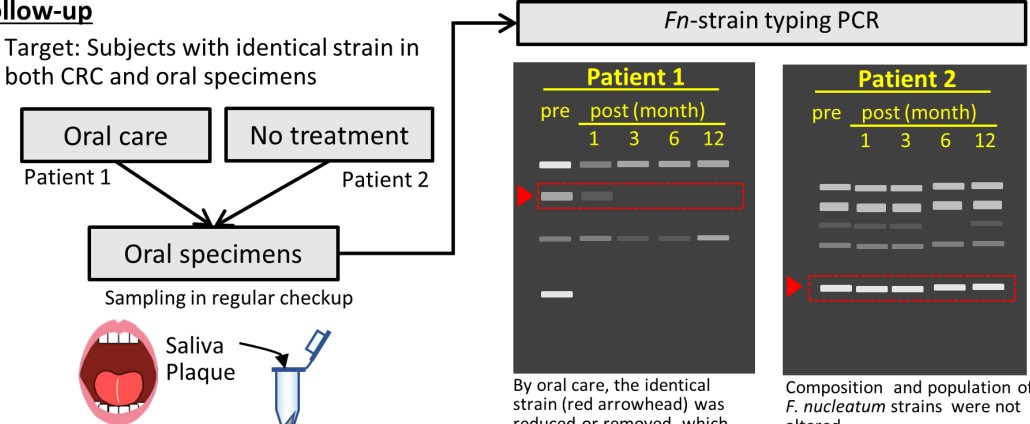

FIG 6 Overview of the CRISPR region-targeted PCR for the identification of *F. nucleatum* at the strain level and its application. (a) Sampling and preservation. Identification of bacteria at the strain level using cryopreserved samples without the need for conventional culture methods. (b) Procedure, interpretation, and application. Total DNA extracted from specimens and purified was analyzed by the *F. nucleatum*-strain genotyping PCR comprising the CRISPR-associated region targeting PCR, and the PCR using a primer set for nested PCR was performed followed by electrophoresis. An amplicon of the same size is found from both CRC and saliva, indicating that the identical strain is present in both CRC and saliva. Using this method, researchers can investigate the percentage of CRC promoted by oral *F. nucleatum* or prevent CRC recurrence. (c) For example, dynamic monitoring of the identical strain in both CRC and saliva, which is likely to be a pathogenic strain, during clinical follow-up. The effect of removal or reduction of this strain in oral cavity can then be investigated.

more sensitive and extensive method than AP-PCR for discriminating *F. nucleatum* at the strain level; however, further studies are warranted to confirm this finding. This method also has low quantifiability and needs to be combined with a quantitative analysis to address its physiological relevance.

In this study, we demonstrated that some strains inhabit both the oral cavity and CRC tissues and developed a CRISPR–Cas region-targeted PCR method for the identification of *F. nucleatum* at the strain level. This method can identify *F. nucleatum* strains in cryopreserved samples and is significantly superior to traditional AP-PCR, which can only be performed on isolates. Therefore, this method eliminates the disadvantages of culturing, such as time loss, skill bias, and low detection limits, and provides sensitive and stable data within a short time for high-throughput analysis. We believe that the new method would help in prevention of recurrence after CRC removal and suppression of carcinogenesis from adenoma.

## MATERIALS AND METHODS

### Searching for CRISPR–Cas system of *F. nucleatum*

The genome sequences of 26 strains of *F. nucleatum* from the NCBI database (https://www.ncbi.nlm.nih.gov/) were downloaded (Table S2); CRISPR sequences were searched using CRISPRfinder program (https://crisprcas.i2bc.paris-saclay.fr/); and Cas genes were identified using the *in silico* Molecular Cloning Genomics Edition (In Silico Biology, Inc.).

### *F. nucleatum* strains used in this study

Type strains, *F. nucleatum* subsp. *animalis* JCM11025$^T$, *F. nucleatum* subsp. *nucleatum* JCM8532$^T$, *F. nucleatum* subsp. *polymorphum* JCM12990$^T$, and *F. nucleatum* subsp. *vincentii* JCM11023$^T$, were purchased from the Japan Collection of Microorganisms of RIKEN BioResource Research Center. Clinical isolates previously identified as *F. nucleatum* obtained from the saliva of five randomly selected CRC patients (Patients D, E, F, G, and M) from our previous study (15) were used. For clinical application, to confirm whether this method can detect identical strains of *F. nucleatum* present in both the CRC tissues and saliva, *F. nucleatum* was isolated from the additional specimen samples (Patients O, P, and Q) following the methods described in our previous paper (15). All *Fusobacterium* strains were incubated on Eggerth-Gagnon agar supplemented with 5% horse blood or GAM Agar, Modified (Nissui Pharmaceutical Co., Ltd., Tokyo, Japan), under anaerobic condition at 37°C for 48 h using AnaeroPack-Anaero (Mitsubishi Gas Chemical Co., Tokyo).

### DNA extraction from bacterial cells and patient samples

The total DNA of *F. nucleatum* strains obtained from our previous study (15) was prepared according to the previously mentioned method (15). The genomic DNA of the strains obtained from Patients D, E, and M was prepared using the Nucleaspin Tissue kit (Takara Bio, Shiga, Japan) according to the manufacturer's instructions.

### DNA preparation from specimen samples

Freshly collected saliva and CRC samples from Patients O, P, and Q and patients with colorectal adenoma and periodontitis were transported and homogenized as previously described (15). The CRC specimen homogenate and saliva samples previously stored at −80°C were thawed on ice for total DNA extraction following a previously described enzymatic lysis method (38).

## Long-read sequencing and bioinformatic analysis

Randomly selected five pairs of isolates obtained from CRC and saliva of each patient with CRC, which were suggested as identical strains by AP-PCR in our previous study (15), were analyzed using whole-genome sequencing [Isolate C-A3 in CRC and Isolate S-A3 in saliva are Strain A3 derived from Patient D; Isolate C-P4 in CRC and Isolate S-P4 in saliva are Strain P4 derived from Patient E; Isolate C-V3 in CRC and Isolate S-V3 in saliva are Strain V3 derived from Patient F; Isolate C-P10 in CRC and Isolate S-P10 in saliva are Strain P10 derived from Patient G; Isolate C-P11 in CRC and Isolate S-P11 in saliva are Strain P11 derived from Patient H in our previous paper (15)]. The genomic DNA of *F. nucleatum* was prepared according to the method previously reported (39). Briefly, bacterial cells were incubated with 15 mg/mL lysozyme (Sigma-Aldrich, St. Louis, MO) and 2,000 U/mL achromopeptidase (Fujifilm Wako, Osaka, Japan) and then treated with 1% sodium dodecyl sulfate (Promega, Madison, WI) and 1 mg/mL proteinase K (Merck Millipore, Darmstadt, Germany). The genomic DNA was extracted with phenol/chloroform/isoamylalcohol (25:24:1), isolated by ethanol precipitation, and purified by RNaseA (Nippongene, Tokyo, Japan) treatment, followed by polyethylene glycol 6000 (Fujifilm Wako) precipitation.

The sequencing was performed with the PacBio Sequel II platform. For the PacBio sequencing, the library (target length, 10 to 15 kbp) was prepared using the SMRTbell template prep kit 2.0 (Pacific Biosciences, Menlo Park, CA). Sequencing data were produced with more than 100-fold coverage and assembled using the assembly program Canu v1.8 (40) to obtain contigs with the size over 2.0 Mb. The quality of the assembled contigs was assessed using CheckM v1.1.3 (41). Accuracy of the assembly was also evaluated by self-mapping of the reads against the assembled contigs.

SNVs between each pair of isolates and between non-paired isolates were analyzed using Snippy version (v4.6.0) (https://github.com/tseemann/snippy).

## Designing of primer sets for detection of the CRISPR-associated region

The CRISPR–Cas system DNA sequences for clinical isolates obtained from our previous study (15) were analyzed after amplification using primers for the CRISPR–Cas system based on DNA sequences of strains in database (Table S2). CRISPR–Cas system sequences of all strains, including clinical isolates and those in the NCBI database, were aligned by Muscle in MEGA7 software; then the target positions and sequences of each primer were determined (Table S3; Fig. S15).

To amplify the CRISPR-associated region (red horizontal line in Fig. 1), a forward primer was designed in the *Cas1* or *Cas2* gene sequence, which is present in all CRISPR–Cas systems, and a reverse primer was designed to a specific gene, which is conserved downstream of CRISPR in all strains. The primer set for the nested-PCR was designed inside the first primer set for CRISPR–Cas system, but the reverse primers for Type I-B1, Type I-B2, and Type III-A were the same as the reverse primer for the CRISPR region (Table S3). In addition, to confirm the presence of the CRISPR–Cas system, primer sets targeting the peripheral region of the CRISPR–Cas system (blue horizontal line in Fig. 1) were designed; briefly, forward primers were designed targeting the upstream gene of the Cas gene present in all strains in the NCBI database, and reverse primers were the same as the reverse primer for the CRISPR region. The annealing temperature (Table S3) was determined based on the Tm value, and amplification was confirmed using genomic DNA obtained from isolates.

## *F. nucleatum*-strain genotyping PCR

In the first round of *F. nucleatum*-strain genotyping PCR, each of the four types of the CRISPR region was amplified by a 30-cycle step using a total volume of 15 µL PCR mixture containing PCR buffer for KOD-Multi & Epi, 0.3 U KOD-Multi & Epi (DNA polymerase) (TOYOBO), 300 nM of each primer set, and 5 ng or 1 µL of genomic/total DNA derived from *F. nucleatum* isolates or specimens, respectively. The thermal cycling condition was

94°C for 2 min; 25 (for bacterial isolates) or 30–40 (for specimen) cycles at 98°C for 10 s, each annealing temperature (Table S3) for 30 s, and 68°C for 3 min and 30 s. In second round of PCR, each of the four types of CRISPR regions was amplified by the 30-cycle PCR under same the condition using the primer sets for nested PCR with 1 μL of 100-fold (bacterial isolates) or 50-fold (specimens) dilution of the first PCR product as the template. The PCR products were electrophoresed on 0.7% (wt/vol) Tris-acetate-EDTA (TAE)-agarose gels and observed with a UV transilluminator Sayaca-Imager (DRC, Tokyo). For the analysis of multiple-strain mixture, equal amounts of each genomic DNA derived from multiple strains were used as a template.

## Sanger sequencing and BLAST analysis

To verify the precision of *F. nucleatum*-strain genotyping PCR, the PCR amplicons were excised from agarose gel and purified using the FastGene Gel/PCR extraction kit (Nippon Genetics, Tokyo, Japan). Both 5′- and 3′-ends of the purified amplicons were Sanger sequenced with the primers used for nested PCR amplification. Sequencing was performed by the FASMAC sequencing service (FASMAC, Kanagawa, Japan). The sequences of the first 29 bases and the dozens of bases at the 3′ end were trimmed according to the sequencing chromatograms to around 600 nucleotides. The trimmed sequences were then analyzed by BLAST search against the nr/nt (for *F. nucleatum* subsp. *nucleatum* ATCC 25586) and the whole-genome shotgun contigs (for the other three type strains) databases with megablast algorithm. The repeat sequences in the obtained sequences were searched using the CRISPRCasFinder online program (https://crisprcas.i2bc.paris-saclay.fr/CrisprCasFinder/Index) and visually confirmed.

## AP-PCR

AP-PCR was performed according to our previous study (15), using primer D11344 (5′-AGTGAATTCGCGGTGAGATGCCA-3′) or D8635 (5′-GAGCGGCCAAAGGGAGCAGAC-3′) (42).

## Sequencing of *F. nucleatum*-strain genotyping PCR amplicons

*F. nucleatum*-strain genotyping PCR amplicons were cloned into the pUC19 cloning vector using In-Fusion Snap Assembly cloning kits (Takara Bio USA, Inc., CA, USA) according to the manufacturer's instructions. A linearized vector with 16 bases homologous to the nested PCR product at both ends was created using the following primers: Type_I-B2_InFusion_F; 5′-GRCAAGTTAGAAAAAAGATCCTCTAGAGTC-GACCTGCA-3′ and Type_I-B2_InFusion_R; 5′-TTTCATTTCTTGATTTGATCCCCGGGTACCGAGCTC-3′. Sequences homologous to the *F. nucleatum*-strain genotyping PCR products are underlined. The sequences of tissue *F. nucleatum*-strain genotyping PCR products and the PCR products from the corresponding isolates were sequenced by the primer walking method.

## Sampling of saliva from patients with colorectal adenoma and periodontitis during oral care

Pre- and post-treatment saliva was collected from patients with both colorectal adenoma and periodontitis receiving oral care for 1–3 months. Five patients were randomly selected from participants. To collect saliva specimen, 10 mL of saline was gargled for 1 min and then collected. This was centrifuged (9,100 × *g* for 3 min), the supernatant was discarded, and the remaining 1 mL of viscous liquid at the bottom was collected as a saliva specimen and cryopreserved at −80°C. Intraoral care and medical guidance were provided by a dentist.

## ACKNOWLEDGMENTS

We thank Dr. Ayano Yamashita of Kyodo Milk Industry Co., Ltd. for supporting this study.

This work was funded by Kyodo Milk Industry Co., Ltd. The funders had no role in the study design, data collection and analysis, decision to publish, or preparation of the manuscript. This work was supported by JSPS KAKENHI, grant numbers 19K18981 to Y. Sugi, 23K17454 to M.M., and 18K07950 to T.H., and Yokohama Foundation for Advancement of Medical Science for Grants for cancer research to T.H.

The authors declare that they have no conflict of interest.

## AUTHOR AFFILIATIONS

[1]Dairy Science and Technology Institute, Kyodo Milk Industry Co. Ltd., Tokyo, Japan

[2]Department of Gastroenterology and Hepatology, Yokohama City University School of Medicine, Yokohama, Japan

[3]Laboratory for Microbiome Sciences, RIKEN Center for Integrative Medical Sciences, Yokohama, Japan

## AUTHOR ORCIDs

Yusuke Ogata ⓘ https://orcid.org/0000-0002-5993-4356
Wataru Suda ⓘ https://orcid.org/0000-0002-2861-9724
Takuma Higurashi ⓘ http://orcid.org/0000-0002-1815-4396
Mitsuharu Matsumoto ⓘ http://orcid.org/0000-0002-0378-3077

## FUNDING

| Funder | Grant(s) | Author(s) |
| --- | --- | --- |
| Kyodo Milk Industry Co., Ltd. | | Mitsuharu Matsumoto |
| JSPS KAKENHI | 18K07950 | Takuma Higurashi |
| Yokohama Foundation for Advanced Medical Science | | Takuma Higurashi |
| JSPS KAKENHI | 19K18981 | Yutaka Sugi |
| JSPS KAKENHI | 23K17454 | Mitsuharu Matsumoto |

## AUTHOR CONTRIBUTIONS

Yumi Shimomura, Conceptualization, Data curation, Formal analysis, Investigation, Methodology, Visualization, Writing – original draft | Yutaka Sugi, Formal analysis, Funding acquisition, Investigation, Methodology, Visualization, Writing – original draft, Data curation | Aiko Kume, Investigation | Wataru Tanaka, Investigation, Visualization | Tsutomu Yoshihara, Resources, Writing – review and editing | Tetsuya Matsuura, Resources, Writing – review and editing | Yasuhiko Komiya, Resources | Yusuke Ogata, Methodology, Writing – review and editing | Wataru Suda, Methodology, Writing – review and editing | Masahira Hattori, Methodology, Writing – review and editing, Supervision | Takuma Higurashi, Conceptualization, Funding acquisition, Resources, Writing – review and editing | Atsushi Nakajima, Writing – review and editing, Supervision | Mitsuharu Matsumoto, Conceptualization, Formal analysis, Funding acquisition, Project administration, Validation, Visualization, Writing – original draft, Writing – review and editing, Data curation

## DATA AVAILABILITY

The complete genome sequences of *F. nucleatum* isolates have been deposited in DDBJ under BioProject accession number PRJDB16243, and the Biosample accession numbers are SAMD00632025 to SAMD00632034. CRISPR–Cas system sequences were deposited in NCBI/ENA/DDBJ under the accession numbers LC583793–LC583797, LC585884–LC585886, and LC592376–LC592400.

## ETHICS APPROVAL

The study protocol complied with the Declaration of Helsinki and the Ethics Guidelines for Clinical Research published by the Ministry of Health, Labour and Welfare of Japan. Approval for this study was obtained from the Ethics Committee of Yokohama City University Hospital. The patients included in the study provided written, informed consent for their participation in the study. The protocol and informed consent forms were approved by the institutional ethics committees at each participating institution.

This study was registered in the UMIN Clinical Trials Registry under ID UMIN000016229.

## ADDITIONAL FILES

The following material is available online.

### Supplemental Material

**Supplemental figures and tables (Spectrum05123-22-S0001.pdf).** Fig. S1 to S15; Tables S1 to S6.

### Open Peer Review

**PEER REVIEW HISTORY (review-history.pdf).** An accounting of the reviewer comments and feedback.

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
