## [Reviewer comments · Microbiology Spectrum]

Microbiology Spectrum

Strain-level detection of *Fusobacterium nucleatum* in colorectal cancer specimens by targeting the CRISPR–Cas region

Yumi Shimomura, Yutaka Sugi, Aiko Kume, Wataru Tanaka, Tsutomu Yoshihara, Tetsuya Matsuura, Yasuhiko Komiya, Yusuke Ogata, Wataru Suda, Masahira Hattori, Takuma Higurashi, Atsushi Nakajima, and Mitsuharu Matsumoto

Corresponding Author(s): Mitsuharu Matsumoto, Kyodo Milk Industry Co. Ltd

Review Timeline:

Submission Date:	December 13, 2022
Editorial Decision:	February 6, 2023
Revision Received:	August 3, 2023
Accepted:	August 25, 2023

Editor: Olga Soutourina

Reviewer(s): Disclosure of reviewer identity is with reference to reviewer comments included in decision letter(s). The following individuals involved in review of your submission have agreed to reveal their identity: Emma Allen-Vercoe (Reviewer #1)

Transaction Report:

DOI: <https://doi.org/10.1128/spectrum.05123-22>

February 6, 2023

Dr. Mitsuharu Matsumoto
Kyodo Milk Industry Co. Ltd
Dairy Science and Technology Institute
20-1 Hirai, Hinode-machi
Nishitama-gun
Tokyo, Tokyo 190-0182
Japan

Re: Spectrum05123-22 (Strain-level detection of *Fusobacterium nucleatum* in colorectal cancer specimens by targeting the CRISPR-Cas region)

Dear Dr. Mitsuharu Matsumoto:

Link Not Available

To respond to the reviewers' criticism, please consider the genome sequencing to validate the results as suggested by reviewer 1. Additional proofreading will be helpful during the revision to improve the quality of the manuscript.

Sincerely,

Olga Soutourina

Journals Department
Reviewer comments:

Reviewer #1 (Comments for the Author):

Fusobacterium nucleatum spp. and close relatives are highly heterogenous. Strain level differences may be critical to disease - and the association of *F. nucleatum* with colorectal cancer has brought this species under increasing scrutiny. Few studies have addressed strain-level differences between *F. nucleatum* strains, in part because this is hampered by the choice of marker

genes used. The manuscript by Ahimomura et al. attempts to remedy this by presenting the use of PCR amplification of gene regions around the CRISPR-Cas regions of strains to easily show strain-level differences through simple PCR and analysing banding patterns using agarose gel electrophoresis.

I am generally very positive about this work as this is a useful approach to take that seem on first glance to be highly differentiating between Fn isolates. I was particularly interested to see the approach used to analyze amplicons obtained from saliva and stool samples of the same patients, as this is a great way to indicate the *F. nucleatum* strains that may be shared between oral and colorectal sites (and help understanding of strains that may have more oncogenic properties than others). I am also pleased to see that the method is sensitive enough to detect strains beyond those that could be isolated. It is unfortunate that this method cannot be used for all strains as not all carry CRISPR-Cas sequences - but for those that do the approach has the potential to open up new areas of enquiry.

I have a few criticisms, detailed below. One major and the rest relatively minor.

MAJOR

While the work is very convincing as presented, that some oral isolates look to be identical to some stool isolates in the same patient, as this work is presented as a novel methodology, this needs to be validated through complete genomic sequencing of these isolates and comparison. This may be a valuable exercise in itself, as the number of accumulated mutations between 2 'identical' isolates may indicate the extent to which isolates travel from one site to another; if this is a rare event that resulted in colonization of the colon from a historical seeding from the oral site, then each pool of clones from each site may be more divergent. If it is more common, there will not be much divergence. Regardless, it is not possible to be able to definitively say that isolates with the same PCR amplicon pattern are clonal (as the authors seem to infer), without careful genomic sequencing. Sequencing bacterial genomes is no longer expensive nor technologically challenging, and can be done as a service by many providers. I strongly suggest that the authors carry out this work on at least 2 example isolates from the same patient, to explore this hypothesis and validate their work further. In my view, this would substantially improve the value of the manuscript to the field.

MINOR

Line 47: Add 'Some tested strains of' before the sentence starting 'This bacterium induce...'

Line 60: 'bacteriology' better than 'bacteria'

Line 107: repetition of 'Type I-B2'

What are the yellow arrowheads in Figure 5a in reference to (please add to the legend).

Reviewer #2 (Comments for the Author):

I reviewed the manuscript entitled "Strain-level detection of *Fusobacterium nucleatum* in colorectal cancer specimens by targeting the CRISPR-Cas region" by Shimomura & Sugi et al. The manuscript is interesting and well written. It describes a new method for detecting strain-level *F. nucleatum* from colorectal cancer and saliva specimens using two PCR reactions to amplify specific CRISPR-Cas regions from *F. nucleatum* (conventional PCR and nested-PCR).

Below are my comments to improve the manuscript:

Line 21: "75% patients" should be "75% of patients".

Line 24: Does "typing" mean "genotyping"?

Lines 60-62: Does this sentence refer to *F. nucleatum*? The authors should mention in this sentence that all these references are related specifically to *F. nucleatum*.

A space is needed between the last word and the references. For example in lines 44, 45, 46, etc. Please revise this throughout the text.

Figure 1 shows that the authors classified CRISPR-Cas region in the DNA of the 26 strains of *F. nucleatum* in Type-IB, II-A, III-A (and two subtypes of Type I-B: Type I-B1, and I-B2), Figure 1 shows that *F. nucleatum* subsp. *animalis* was categorized as Type I-B1, however in Figure 2, the authors show a specific PCR band for *F. nucleatum* subsp. *animalis* as Type I-B1 and Type III-A. Can the authors revisit this part of the manuscript and make sure that all information are described in details so it doesn't raise questions to readers?

The graphs would be easier to follow without the figure legends, if the authors:

- Add the molecular weight for the DNA ladder
- Add on top of the PCR results, which samples are being analyzed and which technique is being used (Saliva vs CRC; F. nucleatum-strain typing PCR vs AP-PCR, etc).

Lines 100-104: The authors mention that they were not able to detect PCR amplicon for F. nucleatum in several cases according to their preliminary study. To overcome this issue, the authors performed a nested PCR to increase CRISPR detectability. If it initially hard for the authors to detect PCR amplicons, does that mean there were very few bacteria in their samples? Can the authors add a discussion to the text on how physiologically relevant the DNA data can be in detecting F. nucleatum in CRC and saliva specimens?

What is the advantage of the method described in this paper compared to the method used in their previous work? Please add this piece of information in the manuscript.

Using this new method describe in the present manuscript, what was the percentage of patients analyzed that showed the same F. nucleatum strain in CRC and saliva samples? Please add this piece of information in the manuscript.

Please add a discussion on the known mechanisms described by F. nucleatum to increases chances of developing CRC, correlation with disease stage, and survival rates.

Staff Comments:

Preparing Revision Guidelines

Please return the manuscript within 60 days; if you cannot complete the modification within this time period, please contact me. If you do not wish to modify the manuscript and prefer to submit it to another journal, please notify me of your decision immediately so that the manuscript may be formally withdrawn from consideration by Microbiology Spectrum.

Reviewer comments:

Reviewer #1 (Comments for the Author):

Fusobacterium nucleatum spp. and close relatives are highly heterogeneous. Strain level differences may be critical to disease - and the association of *F. nucleatum* with colorectal cancer has brought this species under increasing scrutiny. Few studies have addressed strain-level differences between *F. nucleatum* strains, in part because this is hampered by the choice of marker genes used. The manuscript by Ahimomura et al. attempts to remedy this by presenting the use of PCR amplification of gene regions around the CRISPR-Cas regions of strains to easily show strain-level differences through simple PCR and analysing banding patterns using agarose gel electrophoresis.

I am generally very positive about this work as this is a useful approach to take that seem on first glance to be highly differentiating between *Fn* isolates. I was particularly interested to see the approach used to analyze amplicons obtained from saliva and stool samples of the same patients, as this is a great way to indicate the *F. nucleatum* strains that may be shared between oral and colorectal sites (and help understanding of strains that may have more oncogenic properties than others). I am also pleased to see that the method is sensitive enough to detect strains beyond those that could be isolated. It is unfortunate that this method cannot be used for all strains as not all carry CRISPR-Cas sequences - but for those that do the approach has the potential to open up new areas of enquiry.

I have a few criticisms, detailed below. One major and the rest relatively minor.

We thank the reviewer for these valuable comments and suggestions on our manuscript. We have made revisions accordingly.

MAJOR

While the work is very convincing as presented, that some oral isolates look to be identical to some stool isolates in the same patient, as this work is presented as a novel methodology, this needs to be validated through complete genomic sequencing of these isolates and comparison. This may be a valuable exercise in itself, as the number of accumulated mutations between 2 'identical' isolates may indicate the extent to which isolates travel from one site to another; if this is a rare event that resulted in colonization of the colon from a historical seeding from the oral site, then each pool of

clones from each site may be more divergent. If it is more common, there will not be much divergence. Regardless, it is not possible to be able to definitively say that isolates with the same PCR amplicon pattern are clonal (as the authors seem to infer), without careful genomic sequencing. Sequencing bacterial genomes is no longer expensive nor technologically challenging, and can be done as a service by many providers. I strongly suggest that the authors carry out this work on at least 2 example isolates from the same patient, to explore this hypothesis and validate their work further. In my view, this would substantially improve the value of the manuscript to the field.

We thank the reviewer for this suggestion. Accordingly, we analysed five whole-genome sequencing pairs of isolates (ten isolates) obtained from CRC and saliva samples of each patient with CRC, which were suggested to be identical strains by arbitrarily primed polymerase chain reaction (AP-PCR) in our previous study (Gut 68:1335-1337, 2019). Using a pairwise whole-genome comparison for each pair, single nucleotide variants (SNVs) were analyzed. The percentage of SNVs (the number of SNVs/genome size) between paired strains were 0.00036% - 0.0038%, which were significantly lower than those in between non-paired strains (1.228%-3.159%, mean: 2.374%) (see Supplementary Table S1), demonstrating that these pairs of isolates are identical strains. Therefore, we added this information in the revised manuscript (line 23–25 in Abstract, 98–108 in Results, 240–243 in Discussion, line 337–357 in Methods, and Supplementary Table S1 in the revised manuscript).

MINOR

Line 47: Add 'Some tested strains of' before the sentence starting 'This bacterium induce...'

We thank the reviewer for this suggestion. We have revised the manuscript accordingly (line 58 in the revised manuscript).

Line 60: 'bacteriology' better than 'bacteria'

T We thank the reviewer for this suggestion, with which we agree. However, we have deleted this sentence because reviewer 2 commented that the authors should cite references specifically related to *F. nucleatum*, and this reference mentions general pathogenic bacteriology, rather than *F. nucleatum* specifically.

Line 107: repetition of 'Type I-B2'

This sentence indicates the types of CRISPR-associated regions detected by *F. nucleatum*-strain-genotyping PCR from type strains. Because Type I-B2 was detected from both *F. nucleatum* subsp. *nucleatum* JCM8532^T and *F. nucleatum* subsp. *polymorphum* JCM12990^T, this

repetition was not an error.

What are the yellow arrowheads in Figure 5a in reference to (please add to the legend).

We thank the reviewer for this question. We have added a description of the yellow arrowheads in the legend to Figure 5. These two amplicons, which were detected only in the saliva, were also CRISPR-associated regions derived from other *Fusobacterium* strains.

Reviewer #2 (Comments for the Author):

I reviewed the manuscript entitled "Strain-level detection of *Fusobacterium nucleatum* in colorectal cancer specimens by targeting the CRISPR-Cas region" by Shimomura & Sugi et al. The manuscript is interesting and well written. It describes a new method for detecting strain-level *F. nucleatum* from colorectal cancer and saliva specimens using two PCR reactions to amplify specific CRISPR-Cas regions from *F. nucleatum* (conventional PCR and nested-PCR).

Below are my comments to improve the manuscript:

We thank the reviewer for these valuable comments and suggestions on our manuscript. We have made revisions accordingly.

Line 21: "75% patients" should be "75% of patients".

We thank the reviewer for this comment. We have revised the manuscript accordingly (line 21 in the revised manuscript).

Line 24: Does "typing" mean "genotyping"?

Typing means genotyping. Accordingly, we have replaced 'typing' with 'genotyping' throughout the manuscript.

Lines 60-62: Does this sentence refer to *F. nucleatum*? The authors should mention in this sentence that all these references are related specifically to *F. nucleatum*.

This reference mentions general pathogenic bacteriology, rather than specialising in *F. nucleatum*. Therefore, we have deleted this sentence and reference.

A space is needed between the last word and the references. For example in lines 44, 45, 46, etc. Please revise this throughout the text.

As suggested, we have revised this point throughout the manuscript.

Figure 1 shows that the authors classified CRISPR-Cas region in the DNA of the 26 strains of *F. nucleatum* in Type-IB, II-A, III-A (and two subtypes of Type I-B: Type I-B1, and I-B2), Figure 1 shows that *F. nucleatum* subsp. *animalis* was categorized as Type I-B1, however in Figure 2, the authors show a specific PCR band for *F. nucleatum* subsp. *Animalis* as Type I-B1 and Type III-A. Can the authors revisit this part of the manuscript and make sure that all information are described in details so it doesn't raise questions to readers?

We thank the reviewer for this comment. We agree that these figures lack clarity. However, they are correct. Figure 1 provides a schematic representation of the gene arrangement of *F. nucleatum* strains representing each type of CRISPR-Cas system among the strains in the NCBI database, which are not type strains. In contrast, Figure 2 reports the results of the amplification of type strains by *F. nucleatum*-strain-genotyping PCR; i.e. *F. nucleatum* subsp. *animalis* JCM11025^T (Figure 2) has two types of CRISPR-Cas system (Type I-B1 and Type III-A), whereas *F. nucleatum* subsp. *animalis* strain 7_1 has only Type I-B1. Furthermore, the observation that CRISPR-Cas system types differed among strains, even within the same subspecies, is reported in Supplementary Table S2 (in the revised manuscript). Therefore, for clarity, we have added the following comment to the legend of Figure 1: Representative strains are not type strains.

The graphs would be easier to follow without the figure legends, if the authors:

- Add the molecular weight for the DNA ladder

- Add on top of the PCR results, which samples are being analyzed and which technique is being used (Saliva vs CRC; *F. nucleatum*-strain typing PCR vs AP-PCR, etc).

We thank the reviewer for this valuable feedback. As suggested, we have revised the issues in question.

Lines 100-104: The authors mention that they were not able to detect PCR amplicon for *F. nucleatum* in several cases according to their preliminary study. To overcome this issue, the authors performed a nested PCR to increase CRISPR detectability. If it initially hard for the authors to detect PCR amplicons, does that mean there were very few bacteria in their samples? Can the authors add a discussion to the text on how physiologically relevant the DNA data can be in detecting *F. nucleatum* in CRC and saliva specimens?

We thank the reviewer for this comment, which made us realize that we did not explain adequately. Moreover, nested PCR is necessary to increase the specificity of the reaction. If this type of PCR is not performed, there is a risk of detecting a PCR product from the non-CRISPR-associated region of *F. nucleatum*. Therefore, we have added this content in the revised manuscript (line 126–128 in the revised manuscript). In addition, in the Limitations sub-section of the discussion we have added a comment that this method has low quantifiability and needs to be combined with a quantitative analysis to address its physiological relevance (line 293–294 in the revised manuscript).

What is the advantage of the method described in this paper compared to the method used in their previous work? Please add this piece of information in the manuscript.

We thank the reviewer for this valuable comment. We addressed the advantages of this method

compared with AP-PCR throughout the manuscript. Therefore, we have summarised them in the Conclusion section (line 296–300 in the revised manuscript).

Using this new method describe in the present manuscript, what was the percentage of patients analyzed that showed the same *F. nucleatum* strain in CRC and saliva samples? Please add this piece of information in the manuscript.

In this study, we analysed genomic DNA extracted from both CRC and saliva samples from three additional patients with CRC (patient IDs: O, P and Q) using *F. nucleatum*-strain-genotyping PCR, as genomic DNA was not available for both samples in a previous study (Gut 2019). As a result, we detected identical *F. nucleatum* strains in both CRC and saliva samples in 66.7% (2/3) of the patients. However, this number of patients is too small, and further studies are needed. We have added this content in the revised manuscript (line 264–267 in the revised manuscript).

Please add a discussion on the known mechanisms described by *F. nucleatum* to increases chances of developing CRC, correlation with disease stage, and survival rates.

As suggested by the reviewer, we have added this information at the beginning of the discussion (line 222–232 in the revised manuscript).

August 23, 2023

Dr. Mitsuharu Matsumoto
Kyodo Milk Industry Co. Ltd
Dairy Science and Technology Institute
20-1 Hirai, Hinode-machi
Nishitama-gun
Tokyo, Tokyo 190-0182
Japan

Re: Spectrum05123-22R1 (Strain-level detection of *Fusobacterium nucleatum* in colorectal cancer specimens by targeting the CRISPR-Cas region)

Dear Dr. Mitsuharu Matsumoto:

We have now received the comments from the reviewers on your revised manuscript and I am pleased to inform you that your paper has been accepted, and I am forwarding it to the ASM Journals Department for publication. You will be notified when your proofs are ready to be viewed.

Please check that the sequence data deposits to DDBJ database are now open and available upon publication of the article.

Sincerely,

Olga Soutourina
Editor, Microbiology Spectrum
